# A whole blood assay for antibody dependent phagocytosis of Plasmodium falciparum infected erythrocytes

Dilini Rathnayake[1], Wina Hasang[2], Alexander Macpherson[3], HongHua Ding [2], Laurens Manning[4], Moses Laman[5], Maria Ome-Kaius [5], Holger W. Unger[6,7,8], Feiko Ter Kuile [8], Mwayi Madanitsa[9], Bruce Wines [10,11,12], P. Mark Hogarth[10,11,12], Elizabeth H. Aitken [2,13] & Stephen J. Rogerson [1,3] ✉

## Abstract

**Background** Antibodies are used to protect against Plasmodium falciparum malaria. One antibody target, the variant surface antigens, is expressed on infected erythrocytes (IEs). Antibodies to these antigens can either block IE sequestration in the tissues, facilitate natural killer cell-mediated killing, or opsonise IEs for phagocytic clearance by neutrophils and monocytes. **Methods** We developed a high-throughput assay to measure antibody-dependent neutrophil phagocytosis (ADNP) and antibody-dependent cellular phagocytosis (ADCP, by blood monocytes) in the same sample of fresh whole blood. **Results** Here we show that immune plasma mediates ADNP and ADCP in a concentration-dependent manner. Uptake is greater in the presence of complement proteins and is largely dependent on the expression of P. falciparum Erythrocyte Membrane Protein 1 located on the IE surface. Plasma from pregnant Papua New Guinean women with and without placental malaria shows that ADNP and ADCP are associated with protection from placental malaria. ADNP, but not ADCP, using IEs expressing IT4VAR19 (a PfEMP1 variant that binds to endothelial protein C receptor through a DC8 domain cassette) is higher at hospital presentation in children with uncomplicated malaria than in severe malaria. In pregnant women, ADNP and ADCP in whole blood are strongly correlated with one another (Spearman's rho = 0.90), but not with ADNP or ADCP using purified neutrophils and monocytes in the absence of complement proteins. **Conclusions** The whole blood assay is a powerful new tool to assess functional antibodies that may protect against P. falciparum malaria. It allows simultaneous measurement of phagocytosis of opsonised IEs by monocytes and neutrophils.

## Plain language summary

Malaria is a lethal disease spread by mosquitoes that are themselves infected with the malaria parasite Plasmodium falciparum. The immune system uses white blood cells to clear red blood cells that are infected with the malaria parasite. White blood cells clear the infected cells by eating them, and they do this better when the infected cells are coated with two proteins, antibody and complement. To better understand how infected cells are cleared, we developed an assay that measures the number of protein-coated infected cells eaten by white blood cells. When people had high levels of antibody protein, more white blood cells ate the parasites, and this was associated with a lower likelihood of malaria. Overall, this assay can help identify people who are susceptible to, or protected from, malaria, bringing us closer to understanding how the immune system works to eliminate malaria infection.

Malaria is a major cause of illness and death, with 249 million clinical cases and an estimated 608,000 deaths reported in 2022[1]. Over 90% of cases and deaths are due to *Plasmodium falciparum*, and the greatest burden falls on young children and pregnant women. With increasing age or an increasing number of pregnancies, immunity develops and disease burden decreases.

Antibodies form an important component of this immunity, and infusion of immune globulin can treat uncomplicated malaria[2]. Defining the targets and features of protective antibodies has proved challenging because *P. falciparum* is a complex organism with a genome encoding many

thousands of proteins and with several distinct life cycle stages, including merozoites and infected erythrocytes, with distinct responses to each. Uniquely, mature IEs of *P. falciparum* sequester in the vasculature, and this sequestration facilitates parasite growth and survival, but can also lead to end-organ damage and dysfunction[3].

Blood stage infection is responsible for malaria's symptoms, and antibodies to this stage may have multiple functions, including inhibiting invasion of merozoites or sequestration of IEs, and facilitating parasite clearance by innate immune cells through opsonic phagocytosis or

---

activation of immune cells to release inflammatory mediators[4]. Antibodies may cooperate with complement and engage Fc and complement receptors on immune cells. Recent studies confirm important roles for antibodies, complement and neutrophils in the clearance of merozoites and of sporozoites[5–7].

The dominant antigen on the IE surface is *P. falciparum* Erythrocyte Membrane Protein 1 (PfEMP1), a family of high molecular weight variant surface antigens that also mediate IE sequestration[8]. A unique PfEMP1 called VAR2CSA mediates placental sequestration, and evidence suggests that a restricted subset of PfEMP1s that bind to endothelial protein C receptor (EPCR) or EPCR and intercellular adhesion molecule 1 (ICAM-1) cause cerebral malaria[9–11]. Effective antibodies to these PfEMP1 types could block adhesion or facilitate phagocytic clearance of IEs. We recently identified a small number of targets and features of antibody response in pregnant women that were predictive of protection from placental malaria[12]. Amongst these were antibodies that block IE binding to the placental ligand chondroitin sulphate A, antibody-dependent neutrophil phagocytosis (ADNP) of IEs and antibody-dependent cellular phagocytosis (ADCP) of IEs by THP-1 monocyte-like cells.

Whole blood assays allow the simultaneous evaluation of neutrophil and monocyte phagocytosis in the same sample and may better reflect the in vivo situation than assays using purified cells. Small-volume whole blood assays are established tools to measure opsonic phagocytosis of bacteria[13] and indicate the importance of both antibodies and complement in bacterial clearance. A whole blood assay has been used to study ADCP by primary monocytes[14].

We present the development and characterisation of a high-throughput assay using small volumes of whole blood from malaria-naïve donors to quantitate opsonic phagocytosis of IEs by neutrophils and monocytes. Both cell types phagocytose opsonised targets, and antibodies and complement both contribute to phagocytosis, which is partly dependent on PfEMP1 expression. Levels of antibody-induced phagocytosis by neutrophils and/or monocytes are associated with protection from placental malaria or from severe malaria in children.

## Methods
### Human subjects
We have used three sets of samples for conducting our experiments.

### Malaria-exposed pregnant women from Malawi (*n* = 40)
From July 2011 to March 2013, HIV-uninfected pregnant women were enroled at antenatal clinics between 16–28 weeks' gestation in a randomised trial comparing intermittent preventive treatment in pregnancy (IPTp) and Intermittent Screening and Treatment in Malawi (Pan African Clinical Trials Registry ISRCTN69800930)[15]. At enrolment, women's *P. falciparum* infection status was established using light microscopy and polymerase chain reaction (PCR). Blood plasma was separated, stored at −80 °C, and shipped frozen to Melbourne. We randomly selected 10 plasma samples each from primigravid women with and without infection and multigravid women with and without infection (Supplementary Table 1). The study was approved by the Malawian National Health Science Research Committee and the Melbourne Health Human Research Ethics Committee. All participants provided informed written consent, including for their samples to be shipped overseas, and to be used in studies of immunity to malaria.

### Pregnant women with or without placental infection at delivery from Papua New Guinea (*n* = 77)
From November 2009 to August 2012, pregnant women in Madang Province, Papua New Guinea were enroled at antenatal clinics between 14–26 weeks' gestation into a randomised controlled trial of IPTp with sulphadoxine-pyrimethamine (SP) and azithromycin (AZ) or one course of SP and chloroquine (CQ)[16] (ClinicalTrials.gov

NCT01136850). Plasma was collected at enrolment, stored at −80 °C, and shipped frozen to Melbourne. Peripheral blood collected at delivery was tested for *P. falciparum* infection using light microscopy and quantitative PCR and placental biopsies were examined for evidence of placental malaria as described[17]. Plasma samples were selected from women with peripheral parasitaemia, but no IEs in the placenta (non-placental malaria, *n* = 27; all available samples used), and samples from women with active placental infection (IEs in the placenta, *n* = 50, frequency-matched for primigravidity, IPTp regime receipt, bed net use, rural residency, and age), as previously described[12]. Participant characteristics are outlined in Supplementary Table 2. The trial was approved by the PNG Institute of Medical Research Institutional Research Board (IMR IRB 0815), the PNG Medical Research Advisory Council (MRAC 0801), and the Melbourne Health Human Research Ethics Committee (2008.016) and the use of plasma samples was approved by the same bodies (IMR IRB 0922; MRAC 10.50; 2010.017). All participants provided informed written consent including for future use of their samples in studies of immunity to malaria.

### Children
Children aged 0.5–12 years of age presenting to Modilon Hospital in Madang, PNG with severe or uncomplicated malaria were recruited from 2006 to 2009[18]. Children with severe malaria were part of an observational study[18], which included admission to the Paediatric Ward of the hospital, treatment in accordance with PNG National Treatment Guidelines, and collection of clinical data, including detailed neurological examination. Severe malaria was defined based on World Health Organization (WHO) criteria[19] and approximately 55% were males. Children with uncomplicated malaria were recruited from clinics in the same villages as children with severe malaria and were matched by age (within 12 months of the index case) sex and ethnicity and had no features of severe disease. Approximately 55% of children were male. Plasma separated from venous blood collected at enrolment (acute) and approximately eight weeks later (convalescent) from each group was stored at −80 °C and shipped frozen to Melbourne. Written consent was obtained from the child's parent or guardian, including for studies of antibody immunity to IE surface antigens. Participant characteristics are outlined in Supplementary Table 3. All available samples were used. The study was approved by the PNG Institute of Medical Research's Institutional Review Board (IRB No.1103) and the Medical Research Advisory Council of the PNG Department of Health (MRAC 11.12).

### Positive and negative control plasma
Decomplemented (heat-inactivated) plasma samples were pooled from fifteen Malawian pregnant women with high levels of antibody to CS2 IEs (immune plasma or PPS). Pregnant women gave informed consent to participate in a study examining the relationship between malaria antibody and use of IPTp and insecticide treated nets[20] Ethical approval was provided by the Human Research Ethics Committee, Walter and Eliza Hall Institute of Medical Research, Ethical Committee of Pirkanmaa Hospital District in Finland, and the College of Medicine Research and Ethics Committee, University of Malawi. Rabbit immunoglobulins (IgGs) developed against human red blood cells (RaH; Cappel) were used as a commercial positive control at a final concentration of 90 µg/ml. Pooled plasma from malaria-naïve individuals residing in Melbourne (malaria naïve plasma or pooled MC) was obtained from volunteer blood donations from Australian Red Cross Lifeblood (agreement 23-11VIC-13; approved by Melbourne Health Human Research Ethics Committee 2017.319). Donor informed consent included acknowledgement that the blood may be used for research. It was used as the negative control, and phosphate buffered saline (PBS) was used as the no-antibody control to differentiate between opsonic and non-opsonic phagocytosis. All plasmas were used at a final dilution of 1:10.

## Whole blood samples

Heparinised whole blood was collected following informed consent from volunteers, as approved by the Melbourne Health Human Research Ethics Committee.

## Parasite cell lines

Red blood cells (RBCs) infected with the placental binding parasite line CS2 or a CS2 skeleton binding protein 1 knockout[21] or IT4VAR19 expressing the *it4var19 var* gene, a PfEMP1 variant that binds to endothelial protein C receptor through a DC8 domain cassette[22], were cultured as previously described and maintained in a gaseous mixture (5% $CO_2$, 1% $O_2$ and 94% $N_2$) in an incubator at 37 °C[23]. The IT4VAR19 EPCR-binding parasite isolate was regularly phenotyped for its expression of the specific PfEMP1 (or DC8[24] using antiserum against IT4VAR19-IEs developed in rabbits[25]. The cultures were synchronized by sorbitol treatment[26] and regularly selected for knob expression by gelatin flotation[27]. Cell cultures were tested for mycoplasma negativity (MycoAlert Kit, Lonza, Mount Waverley, Australia) as per manufacturer's instructions). Mature trophozoite stage IEs were isolated by Percoll® density gradient centrifugation (GE Healthcare USA)[28]. Purity by microscopy was at least 95%. Purified IEs were labelled with dihydroethidium (DHE) (Sigma Aldrich) 25 µg/ml diluted 1:400 for 30 min in the dark at RT and washed to remove excess DHE. The final concentration of IEs was adjusted to $1.65 \times 10^7$/ml.

## Whole blood

Human blood was collected from healthy malaria-naïve adult volunteers from Melbourne into lithium heparin tubes (Beckton, Dickinson and Company, USA), diluted 1:1 with prewarmed RPMI-1640 (Gibco) and plated at 25 µl/well in duplicate in 96-well U-bottom plates (Corning).

## Antibody-Dependent Phagocytosis by neutrophils and monocytes

Ninety-six well U bottom plates were precoated with 1% bovine serum albumin (BSA) (Sigma Aldrich) in PBS at RT for one hour. The coating solution was discarded, and heat-inactivated human plasma samples were added for opsonisation along with 30 µl of DHE-labelled IEs (concentration $1.65 \times 10^7$/ml). The plates were gently tapped to mix the IEs with plasma and incubated in the dark for one hour at RT. The IEs were then washed three times with RPMI-1640 with 25 mM HEPES and centrifuged at 350 x g for 5 min at RT. Opsonised IEs were resuspended in 50 µl of RPMI-1640 and 25 µl/well were added to the diluted human blood prepared in a separate 96-well U bottom plate to reach a final dilution of 1:4 for whole human blood. The neutrophils and monocytes in blood were allowed to phagocytise opsonised IEs at 37 °C in a 5% $CO_2$ incubator for one hour. Phagocytosis was stopped by centrifugation at 350 x *g* for five minutes at 4 °C.

The cell pellets were labelled with fluorescent antibodies against leucocyte cell membrane markers. These included 1:100 FITC antihuman CD14 (BioLegend Cat no. 301804), 1:200 PECy7 antihuman CD16 (BD Biosciences cat no. 557744), 1:800 BV421 antihuman CD66b (Beckton Dickinson Cat no. 562940), and 1:100 FITC antihuman CD45 (BD Biosciences cat no.11-0459-42). All antibodies were diluted in cold RPMI-1640, and 50 µl of the antibody cocktail was added per well. Following incubation on ice in the dark for 30 min, the plates were spun at 350 x *g* for 5 min at 4 °C and the supernatants were discarded. Erythrocytes were lysed by adding 100 µl of 1X FACS lysing solution (BD BioSciences, cat no. 89882) diluted in sterile distilled water at 4 °C for 10 min. The lysis step was repeated once to enrich the leucocyte population and facilitate cell acquisition. Upon lysis, the plates were fixed with cold 2% paraformaldehyde (PFA) in PBS for 10–15 min at RT. The plates were washed (350 x g, five minutes at RT) and resuspended in cold FACS buffer (containing 2% FBS, 2 mM EDTA, and 0.02% sodium azide in PBS) until acquired by a flow cytometer.

Phagocytosis was measured by gating on neutrophils and monocytes using forward scatter (FSC) and side scatter (SSC) parameters. All samples were run in duplicate, and duplicates were averaged. Assays were repeated with three different whole blood donors. For each donor, the percentages of neutrophils or monocytes positive for DHE in the duplicates were averaged. The mean percentage phagocytosis of the three donors for each cell type was used.

## Flow cytometry

We utilised the differences in light scatter properties of RBCs and leucocytes under blue SSC-A (488 nm) and violet light SSC-A (405 nm) to separate the cell types. Leucocytes were identified using fluorescently labelled antibodies against CD45 (leucocyte common antigen), and specific populations were identified using the fluorescent antibodies described above.

## Image stream analysis

DHE-labelled IEs opsonised with immune plasma, malaria naive plasma, or unopsonised IEs were incubated with prediluted whole blood. Following phagocytosis, the RBCs were lysed, and the leucocytes were labelled with fluorescent antibodies against cell surface markers for neutrophils and monocytes as described above. Two thousand five hundred leucocytes in focus were acquired using an Amnis ImageStream flow cytometer (Amnis) based on CD45 positivity. CD45 was used as a standard leucocyte membrane marker to demonstrate whether DHE-labelled parasites were located within leucocytes.

## Complement and complement inhibition

To inhibit the complement pathway, we used compstatin, a potent C3 inhibitor that selectively inhibits C3 activity of complement cascade[29] and a lipocalin protein, OmCI (*Ornithodoros moubata* Complement Inhibitory protein, 1 µM), that specifically binds to C5, inhibiting the downstream complement cascade[30].

Whole blood was prediluted at a volume ratio of 1:1 in RPMI-1640 and compstatin was used at a range of concentrations from 200 µM-1.6 µM. Whole blood was incubated at RT for 15 min with compstatin 200 µM or 1 µM OmCI before being used in ADP assays.

## Stimulation of whole blood for phagocytosis

Two stimulants were used, TNFα and C5a. For TNFα, we used ten-fold dilutions from 10 ng/ml to 0.1 ng/ml, while for C5a, we used concentrations ranging from 1–500 nM at semi-log intervals. After selecting an optimal concentration of C5a and TNFα for whole blood priming, we stimulated prediluted whole blood with either C5a, TNFα, or both C5a and TNFα together to investigate the effect of whole blood priming on ADP of IEs. The stimulations were carried out at RT.

## ELISA for antibody to VAR2CSA

Antibody to full-length VAR2CSA (gift from Morten Nielsen and Ali Salanti, University of Copenhagen) was measured by ELISA as described[12] with the following modifications. Plates were coated with VAR2CSA at a final concentration of 0.5µg/mL in PBS. Plasma samples were diluted 1/200. The detection antibody was goat anti-human IgG HRP (Millipore AP112P). Antibody levels were expressed as arbitrary units, calculated from raw absorbance readings derived against a standard curve made using serial doubling dilutions of PPS starting at a dilution of 1 in 250.

## Statistics and reproducibility

The data obtained from flow cytometry were analysed using FlowJo and exported to Microsoft Excel. The analyses were conducted, and graphs were generated in GraphPad Prism.

All phagocytosis assays were run in duplicate, and the percentages of neutrophils or monocytes positive for DHE in the duplicates were averaged. Assays were repeated with two or three different whole blood donors. For comparisons of clinical samples, three whole blood donors were used and the mean percentage phagocytosis for each cell type was obtained. Phagocytosis levels by groups were compared using the Mann-Whitney U test. For paired samples, the Wilcoxon matched pairs rank

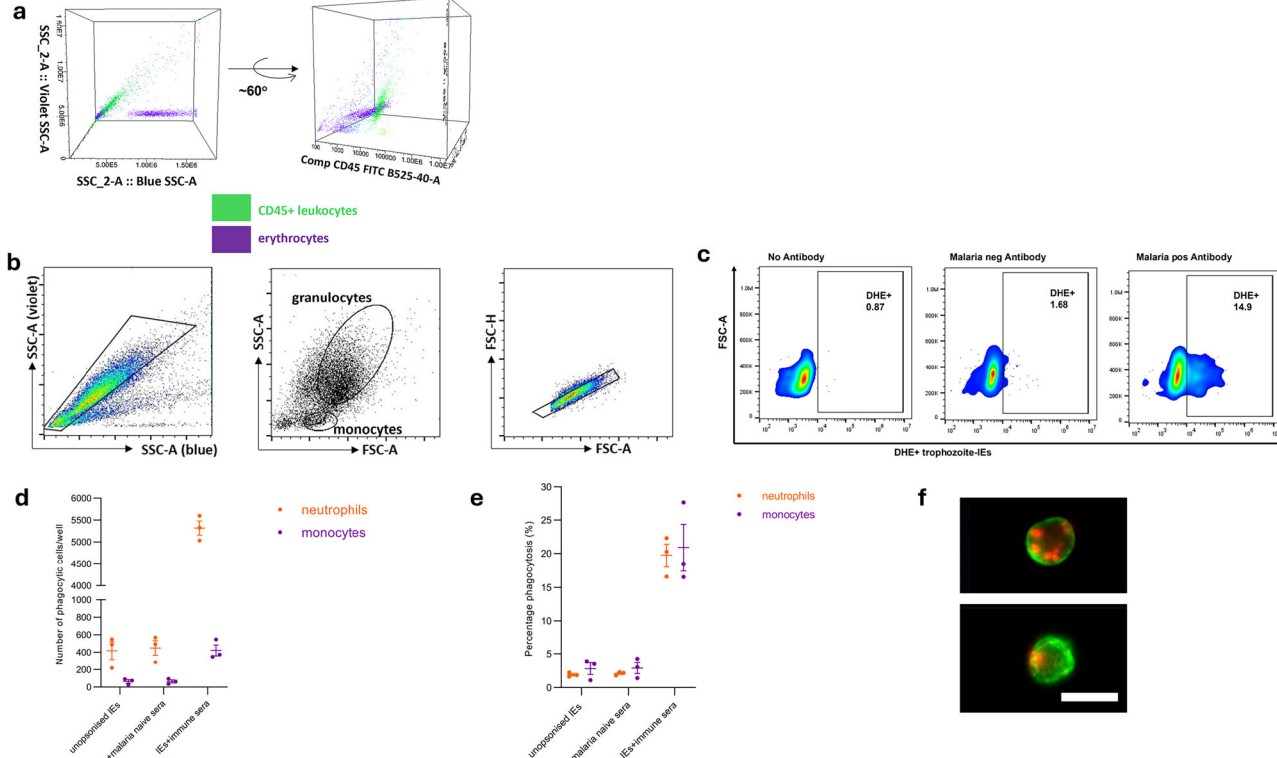

**Fig. 1 | Diagrammatic representation of phagocytosis of *P. falciparum*-IEs in the presence and absence of malaria-specific antibodies. a** Identification of leucocytes in whole human blood using differences in light scatter properties. The three-dimensional view of unlysed whole blood when visualised through the 405 nm violet (y-axis) and 488 nm blue side scatter (SSC-A) shows two prominent populations, the leucocytes (in green) and the erythrocytes (in purple) lying in parallel to blue SSC-A. **b** The granulocyte and monocyte populations were separately identified by the differences in their side (SSC-A) and forward (FSC-A) scatter, and confirmed using cell surface markers, anti-CD66b conjugated to brilliant violet 421 (BV421) and anti-CD16 conjugated to phycoerythrin-Cy7 (PECy7) for neutrophils and anti-CD14 conjugated to fluorescein isothiocyanate (FITC) and anti-CD16 conjugated to PECy7 for monocytes respectively. The doublet granulocytes and monocytes were excluded using a single cell gating. **c** The neutrophils or monocytes positive for dihydroethidium (DHE)-labelled IEs were identified with respect to no antibody (unopsonised) control. Malaria neg Antibody: pooled malaria-naïve plasma, Malaria pos Antibody: pooled immune plasma (PPS) from malaria-exposed individuals **d** Numbers of neutrophils (orange) and monocytes (purple) involved in IE phagocytosis. **e** Percentages of neutrophils or monocytes in the sample that are DHE-positive. In **d** and **e**, three different whole blood donors were tested and dots represent the mean of duplicates from individual experiments, horizontal lines and error bars represent the mean and standard error of the mean. **f** represents the localisation of IEs within neutrophils and monocytes in whole blood when malaria antibodies are present by imaging flow cytometry. Green, CD45 staining of leucocytes; red, DHE staining of IEs. The scale bar represents 15 μm.

test was used. Spearman's correlation was used to compare phagocytosis between donors.

### Reporting summary

Further information on research design is available in the Nature Portfolio Reporting Summary linked to this article.

## Results

### Establishing a whole blood assay to measure opsonic phagocytosis of *P. falciparum*-IEs by both neutrophils and monocytes

This novel flow cytometry-based assay utilises the differences in light scatter properties of RBCs and leucocytes. Due to their haemoglobin content, RBC readily absorb violet (405 nm) light, while the leucocytes do not, giving them distinct side scatter properties (Fig. 1a). Using a conventional flow cytometer equipped with lasers for both blue (488 nm) and violet (405 nm) side scatter (known as blue SSC-A and violet SSC-A hereafter), we show that leucocytes can be distinguished from RBCs, and can be identified by using fluorescently labelled antibodies against a leucocyte specific marker, CD45 (Fig. 1a). Given that RBCs outnumber leucocytes approximately 1000 times in whole blood, RBC lysis was required to permit the timely acquisition of data on a statistically meaningful number of leucocytes. Following whole blood lysis and fixation, leucocyte acquisition was increased from 50% to 75%

relative to non lysed non-fixed whole blood. Proportions of neutrophils and monocytes phagocytosing opsonised and unopsonised IEs were similar in blood with and without lysis and fixation (Supplementary Fig. 1).

The neutrophils and monocytes were identified by their FSC and SSC light scatter properties (Fig. 1b) and confirmed using specific cell-surface markers; CD16 and CD66b for neutrophils and CD14 and CD16 for monocytes. The proportion of leucocytes that phagocytose IE opsonised with immune plasma was substantially (14.9%) greater than that for IE incubated with malaria-naïve plasma (1.68%) (Fig. 1c). The total number of phagocytosing cells was higher for neutrophils than monocytes, reflecting their higher numbers in peripheral blood (Fig. 1d). In the presence of immune plasma, the mean phagocytosis of IEs by both neutrophils and monocytes was approximately 10-fold higher than that for IEs opsonised with malaria naive plasma and unopsonised IEs, and the proportions of neutrophils (mean ± standard error of mean, SEM, 19.7 ± 2.9%) and monocytes (mean ± SEM 20.9 ± 6.0%) taking up IE were similar (Fig. 1e). To ensure that the fluorescent signals measured as phagocytosis by flow cytometry represented internalised IEs and not IEs bound to the cell surface, we used imaging flow cytometry to capture single leucocytes in focus, confirming that IE were internalised by both neutrophils and monocytes (Fig. 1f; Supplementary Fig. 2). IEs opsonised with immune plasma were internalised at rates that were

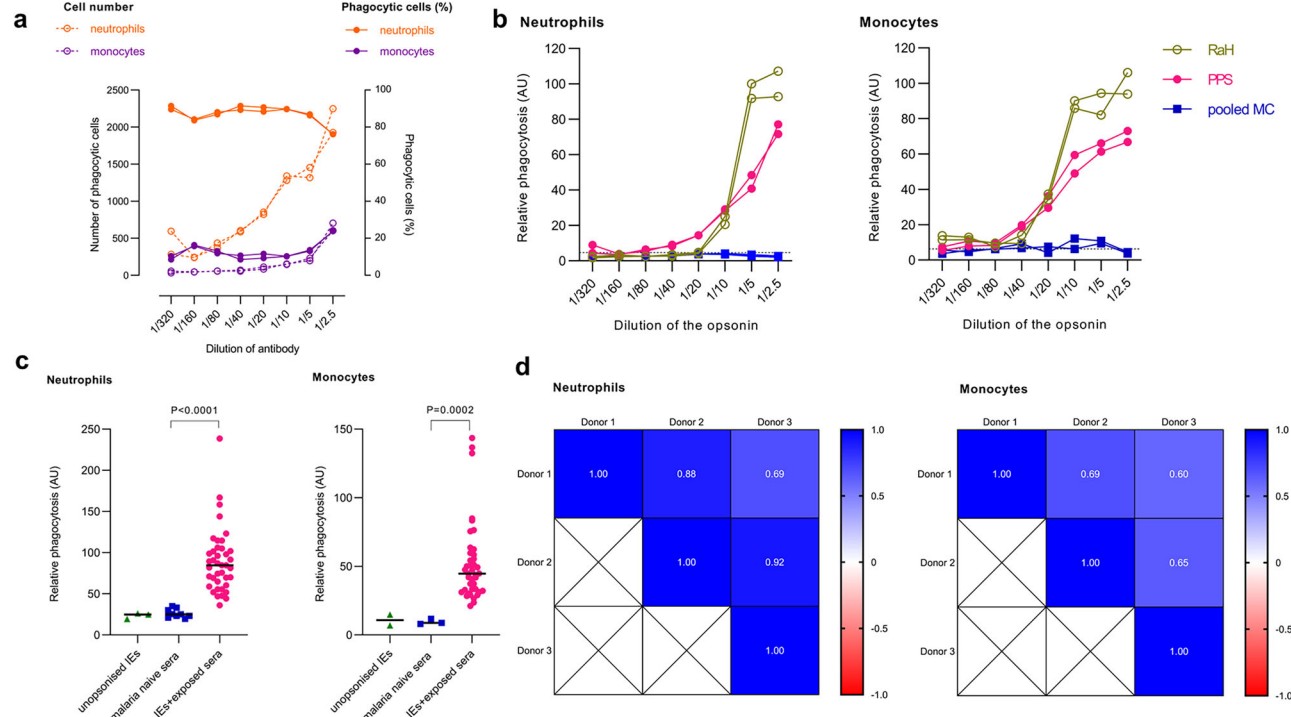

**Fig. 2 | The degree of antibody opsonisation affects antibody-dependent pha-gocytosis (ADP) of *P. falciparum* IEs by neutrophils and monocytes in whole blood. a** Right y-axis, the proportion of phagocytosing cells that are neu-trophils or monocytes (continuous lines and dark circles) over different con-centrations of immune plasma (PPS) Left y-axis, the number of DHE+ neutrophils or monocytes per well (dotted lines and open circles). **b** Concentration-dependent phagocytosis of IEs opsonised by pooled patient plasma (PPS); rabbit anti-human erythrocyte antibody (RaH), maximum concentration 90 µg/ml; and malaria-naïve plasma (pooled MC), expressed relative to RaH 90 µg/ml in arbitrary units (AU). Horizontal dotted line shows non-opsonic phagocytosis. **c** ADP of *P. falciparum*-IEs

by neutrophils and monocytes in whole blood is exposure-specific. The phagocytosis of IEs opsonised with plasma from 40 pregnant Malawian women (malaria-exposed) and from nine (neutrophils) or three (monocytes) malaria-naïve individuals from Melbourne was calculated relative to PPS and expressed in AU. Dots represent the means of three independent experiments each in duplicates using different whole blood donors; black lines indicate median values. Groups compared using Mann-Whitney U Test. **d** Spearman's correlation between levels of phagocytosis observed using three different donors for the 40 malaria-exposed samples shown in **c**. **a**, **b** Mean and standard error of the mean (SEM) from two independent experiments.

substantially greater than for IEs opsonised with non-immune plasma, Supplementary Fig. 2. We confirmed (Supplementary Fig. 3) that monocyte numbers were not selectively depleted by adhesion to plates in these experiments.

## Neutrophil and monocyte phagocytosis of IEs is dependent on the concentration of opsonising antibody

To examine the effect of antibody concentration on opsonic phagocytosis of IEs by neutrophils and monocytes in whole blood, we opsonised IEs with increasing dilutions of immune plasma (Fig. 2a). More neutrophils than monocytes phagocytosed IEs, in keeping with the predominance of the former in whole blood (Fig. 2a). Phagocytosis was dependent on plasma concentration, and the dilution of antibody that triggered at least half the maximum observed was between 1/2.5 to 1/5 dilution for neu-trophils while for monocytes the dilution was between 1/5 to 1/10. At high concentrations of antibody, both neutrophils and monocytes in whole blood were actively engaged in IE phagocytosis further confirming our findings. To determine the optimal dilutions of opsonins for ADP in whole blood, DHE-labelled IEs were opsonised with increasing dilutions of RaH, PPS, or pooled MC, and co-incubated with prediluted whole blood (Fig. 2b). The percentage phagocytosis of IEs opsonised with RaH or PPS by neutrophils and monocytes increased in a concentration-dependent manner, whereas phagocytosis of IE opsonised by non-immune MC plasma remained low even at the highest concentrations. For RaH, phagocytosis by both neutrophils and monocytes appeared to reach saturating concentrations.

Next, we investigated the ability of plasma from individual malaria-exposed pregnant women to promote opsonic phagocytosis of *P. falci-parum*-IEs by both neutrophils and monocytes in whole blood (Fig. 2c). Opsonisation of IEs with plasma samples from pregnant Malawian women[15] resulted in significantly greater uptake of CS2 IEs by neutrophils and monocytes than opsonisation with Melbourne control plasmas ($P < 0.0001$, $P = 0.0002$ respectively). Neutrophil and monocyte phagocy-tosis of IE opsonised with plasma from Malawian women was moderately correlated (Supplementary Fig. 4). Interassay correlation was generally stronger for different neutrophil donors than for different monocyte donors (Fig. 2d), possibly reflecting the substantially greater numbers of neu-trophils assessed.

## The inhibition of complement activation decreases neutrophil and monocyte phagocytosis of antibody-opsonised IEs in whole blood

We used the complement blockers compstatin (a potent and selective C3 inhibitor[29]) and OmCI (*Ornithodoros moubata* Complement Inhibitory protein) which binds to C5 inhibiting the downstream complement cascade[30]. Serial dilutions of compstatin were performed (Supplementary Fig. 5). The concentration of OmCI for optimal complement inhibition was previously defined[30].

Compstatin decreased ADP of immune sera opsonised by neutrophils and monocytes by 46 and 24% respectively (Fig. 3). Addition of OmCI reduced both neutrophil and monocyte ADP of immune sera opsonised IE by about 50% (Fig. 3).

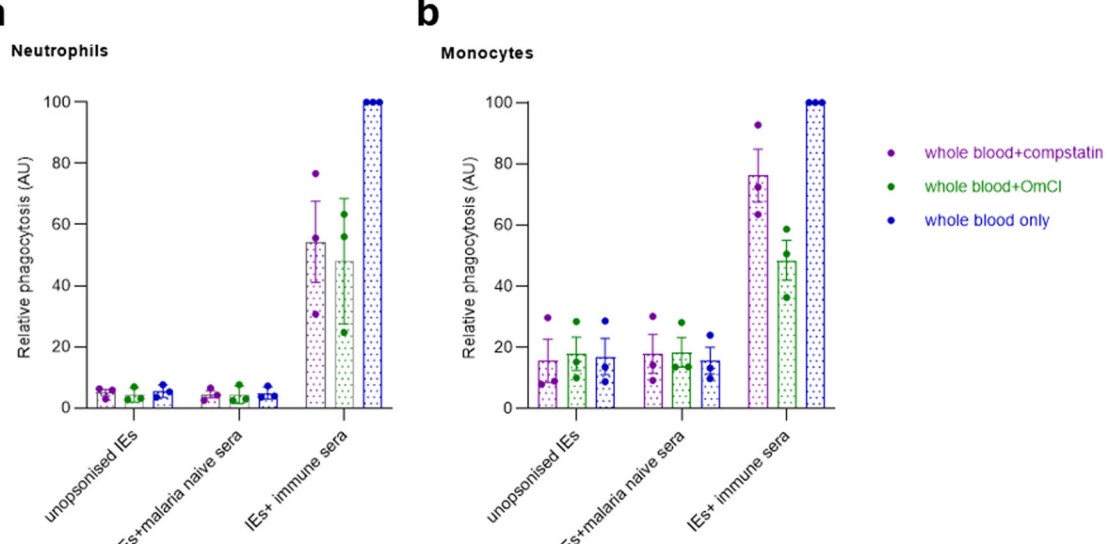

**Fig. 3 | The inhibition of complement activation decreases neutrophil and monocyte phagocytosis of antibody-opsonised IEs in whole blood.** The effect of complement inhibition on ADP of *P. falciparum*-IEs by neutrophils (**a**) and monocytes (**b**) in whole blood is plotted. The y-axes show phagocytosis in arbitrary units (AU) relative to the highest percentage phagocytosis of the positive control opsonin, rabbit anti-human erythrocyte antibody (RaH) in unmodified whole blood. Compstatin (200 µM) and OmCI (50 µM) were used as complement blockers. Bars represent means and whiskers represent the standard error of the mean of three independent experiments, each conducted in duplicates, and each point represents the mean from one experiment.

### Inflammatory mediators released by immune activation enhance ADP of antibody-opsonised *P. falciparum*-IEs

We used two potent inflammatory mediators, TNFα and C5a. We first determined the effects of varying concentrations of TNFα and C5a on phagocytosis of IEs (Fig. 4a). There were concentration-dependent increases in relative phagocytosis of *P. falciparum*-IEs. At the highest concentration of TNFα (10 ng/ml), ADP by neutrophils increased two-fold, whereas for monocytes, ADP increased approximately 1.25-fold (Fig. 4a). When whole blood was stimulated with C5a, ADP of IEs was enhanced about 1.6-fold at the highest concentration of C5a (500 nM) for both neutrophils and monocytes compared to unstimulated whole blood.

When TNFα (10 ng/ml) and C5a (25 nM) were added separately or in combination (Fig. 4b), TNFα increased ADP of IEs by neutrophils by 99% and monocytes showed a 59% increase compared to unstimulated whole blood (Fig. 4b). When stimulated with C5a, ADP increased by 71% for neutrophils and 69% for monocytes, and when TNFα and C5a were combined, ADP increased by 148% for neutrophils and by 77% for monocytes compared to unstimulated whole blood (Fig. 4b).

### PfEMP1 is a major target for antibodies that promote phagocytosis of *P. falciparum*-IEs by neutrophils and monocytes in whole blood

Given that PfEMP1 is implicated as a primary target of antibodies against *P. falciparum* IEs[8], we investigated its role as a target of antibodies promoting ADNP and ADCP in whole blood. Using a flow cytometry-based variant surface antigen recognition assay, immune plasma gave high recognition of placental-like parasite line CS2, and negligible recognition of a CS2-SBP1 transgenic parasite line that lacks functional PfEMP1 on the cell surface[21] (Fig. 5a).

Phagocytosis of opsonised CS2 SBP1KO-IEs was only 56% lower than CS2 IEs for neutrophils, and 48% lower for monocytes (Fig. 5b). We hypothesised that there might be secondary targets of antibodies that immune plasma of pregnant women can identify on the IE surface. To test this hypothesis, we treated uninfected erythrocytes with the RBC aging agent BS3 (bis(sulphosuccinimidyl)suberate)[31], or left them untreated (Fig. 5c). Neutrophil phagocytosis of RBCs was almost entirely dependent on antibodies, whereas 5–10% of monocytes demonstrated non-opsonic phagocytosis in the absence of antibodies. The treatment of normal RBCs with BS3 increases erythrophagocytosis by both neutrophils and monocytes in the presence of antibodies only, by about 20%.

### In *P. falciparum*-infected pregnant women, opsonising antibodies against placental binding CS2 IEs are associated with protection from placental malaria

Seventy-seven plasma samples collected in second trimester from malaria-exposed pregnant women from PNG with and without placental malaria at delivery (Supplementary Table 2) were heat-inactivated and used to opsonise CS2 IEs. ADP of CS2-IEs by both neutrophils ($P < 0.0001$) and monocytes ($P = 0.0007$) was significantly higher in malaria-infected pregnant women with no placental malaria compared to those with placental malaria at delivery (Figs. 6a, b). Both neutrophil and monocyte phagocytosis of IEs showed moderate to strong correlation among three different effector cell donors (neutrophils, r = 0.69–0.92; monocytes, r = 0.60–0.69). Neutrophil and monocyte phagocytosis using whole blood correlated with one another but not with ADNP or ADCP using purified cells (Supplementary Table 3). Neither neutrophil nor monocyte phagocytosis correlated with IgG to full-length VAR2CSA measured by ELISA (Supplementary Fig. 6).

### Antibodies that opsonise IEs for phagocytosis by neutrophils and monocytes in children with severe or uncomplicated malaria

IT4VAR19 IEs were opsonised with plasma samples from children from PNG who had severe or uncomplicated malaria (Supplementary Table 4). Using acute samples collected at admission, ADNP was significantly higher in uncomplicated malaria than severe malaria ($P < 0.0001$) (Fig. 7a). Between admission and convalescence, ADNP of IT4VAR19 IEs declined, more markedly in uncomplicated ($P < 0.0001$) than severe malaria ($P = 0.11$), and the two groups did not differ significantly in convalescence. When the analysis was restricted to paired acute and convalescent samples from 43 children with uncomplicated malaria and 75 children with severe malaria, both groups showed a significant decrease in antibody from presentation to convalescence (Fig. 7b).

At presentation, ADCP did not differ between plasma from severe and uncomplicated malaria, but using convalescent plasma, ADCP was higher in uncomplicated malaria than severe malaria ($P < 0.0001$) (Fig. 7c).

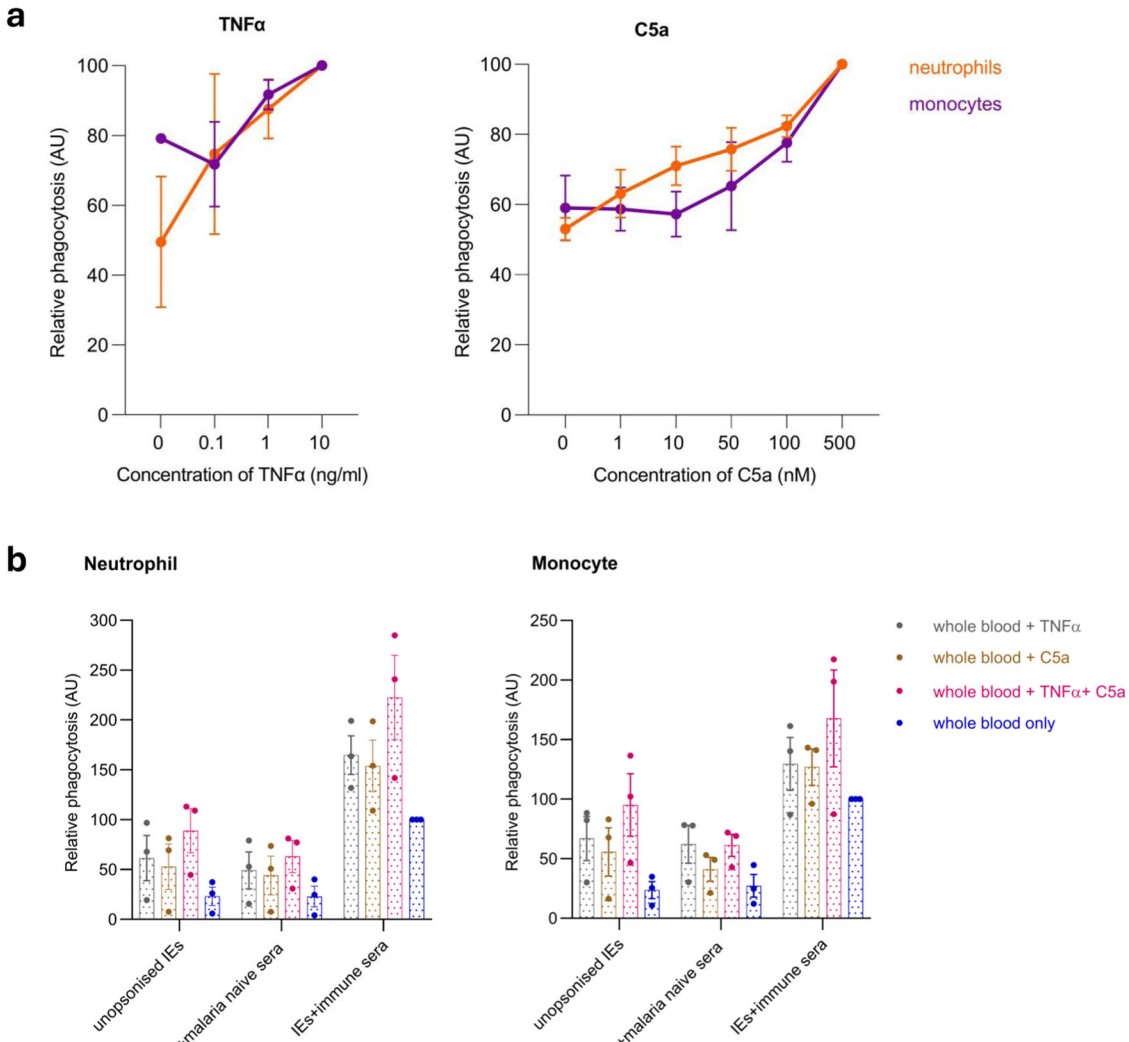

**Fig. 4 | Inflammatory mediators released by immune activation enhance ADP of antibody-opsonised *P. falciparum*-IEs. a** Dose-response curves for relative phagocytosis of antibody-opsonised *P. falciparum*-IEs following stimulation by tumour necrosis factor alpha (TNFα) and complement component C5a respectively. For TNFα, stimulation was carried out for 30 min at RT. For C5a, stimulation was carried out for 15 min at RT. Neutrophil (orange) and monocyte (purple) phagocytosis of IE following exposure to varying doses of TNF or C5a are shown relative to pooled positive plasma (PPS). **b** Neutrophil and monocyte phagocytosis of IEs in whole blood stimulated with TNFα (10 ng/ml), C5a (25 nM), or both, relative to unstimulated whole blood. Opsonisation by pooled immune plasma or pooled malaria-naïve plasma is compared to unopsonised IEs. The results are expressed relative to phagocytosis of IEs in unstimulated whole blood. Mean and standard error of the mean from three independent experiments each performed in duplicate.

Between presentation and convalescence, levels of opsonising antibody to IT4VAR19-IEs declined significantly in severe malaria ($P < 0.0001$) but not in uncomplicated malaria ($P = 0.76$). Analysing paired samples, plasma from children with severe malaria with acute infection promoted substantially more ADP by monocytes than convalescent samples from the same children ($P < 0.0001$), whereas paired samples from children with uncomplicated malaria showed no significant differences between presentation and convalescence (Fig. 7d).

## Discussion

Correlates of immunity against blood-stage malaria are challenging to identify, in part due to a limited understanding of antibody functions in malaria. We describe a whole blood assay that quantitates the opsonic phagocytosis of IEs by neutrophils (ADNP) and monocytes (ADCP). ADNP and ADCP were dependent on the concentration of immune plasma and on PfEMP1 expression on IEs and were enhanced by complement and by leucocyte activation. ADNP and ADCP using CS2 IEs correlated with protection against placental malaria, and ADNP using

IT4VAR19 IEs correlated with protection against severe malaria in young children.

Most previous studies of antibody-dependent responses in malaria have used isolated leucocyte fractions, such as peripheral blood leucocytes[6], or have purified specific leucocyte types such as monocytes, natural killer (NK) cells or neutrophils, using positive or negative selection, or specific forms of density gradient for neutrophils[6,32,33]. These approaches can yield highly purified populations, but they remove the cells from their original milieu, depleting factors including complement and potentially altering their activity. We used cellular characteristics and specific markers to identify the neutrophil and monocyte populations in whole blood with minimal dilution, and then used osmotic lysis to deplete RBCs and increase the speed of acquisition of leucocytes, facilitating the development of a high-throughput assay.

Using specific markers, we identified neutrophils and monocytes and showed that both cell types are engaged in phagocytosis of IEs. Recent studies using purified cells have quantitated the uptake of merozoites or sporozoites by purified neutrophils and monocytes and find that

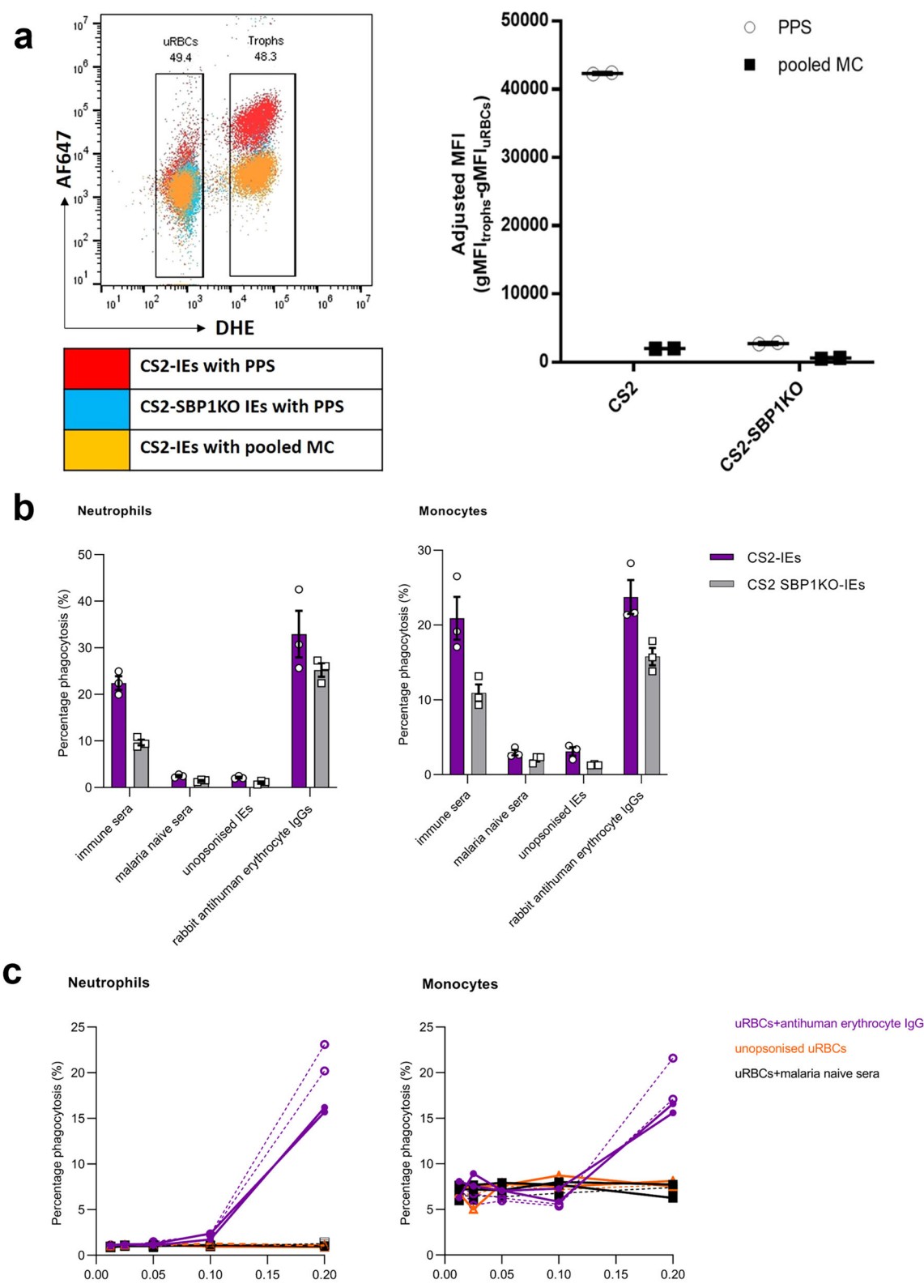

neutrophils were more actively engaged in opsonic phagocytosis than monocytes, at least at higher plasma dilutions[5,6]. By contrast, similar proportions of monocytes and neutrophils ingested IEs.

Complement-fixing antibodies to CS2 IEs have been associated with protection from placental malaria[34], and complement was critical for opsonic phagocytosis of IEs by intermediate monocytes in whole blood[14]. We

confirmed this important role in ADCP and showed that complement is also critical for ADNP, as the complement inhibitors compstatin and OmCI both significantly reduced ADNP and ADCP.

Complement activation leads to the generation of C5a, which can in turn, increase expression of FcγRIII and decrease inhibitory FcγRIIb expression[35]. The inflammatory cytokine TNFα also increases FcγRIIIb

**Fig. 5 | PfEMP1 is a major target for antibodies that promote phagocytosis of *P. falciparum* IEs by neutrophils and monocytes in whole blood. a** Flow cytometry-derived overlay of recognition of DHE-labelled parasite line CS2 and CS2 SBP1KO (skeleton binding protein 1 knockout) IEs by immune plasma (PPS, pooled positive plasma) and non-immune plasma (Melbourne controls, MC). IEs were labelled with dihydroethidium (DHE) and the human antibodies binding to IEs were recognised by an anti-human secondary antibody conjugated to Alexa Fluor 647 (AF647). The binding of IEs was given as an adjusted median fluorescent intensity (MFI) value, which is the geometric MFI (gMFI) signal from trophozoite-stage parasites (trophs), minus the gMFI signal from uninfected red blood cells (uRBCs). Graphs show the adjusted MFI for CS2 and CS2 SBP1KO-IEs opsonised with either immune plasma (PPS) or pooled malaria naïve plasma (pooled MC) from a single experiment carried out in duplicate. **b** ADP of CS2 and CS2 SBP1KO-IEs by neutrophils and monocytes

in whole blood. Phagocytosis was expressed as the mean and standard error of the mean (SEM) of three individual whole blood donors run in duplicates. Three opsonins were used, namely, immune plasma, malaria naive plasma, and rabbit antihuman erythrocyte IgGs, along with an unopsonised IE control. **c** The effect of artificial ageing of uRBCs on phagocytosis. uRBCs were treated with 1 mM BS3 for 15 min at 37 °C. uRBCs were opsonised with rabbit antihuman erythrocyte IgGs (purple solid or dashed lines, open and closed circles), or pooled malaria naive plasma (black solid or dashed lines, open and closed squares), or were unopsonised (red solid or dashed lines, open and closed triangles). The solid lines represent untreated uRBCs, while the dashed lines represent uRBCs treated with BS3. The results are expressed as mean and SEM from two whole blood donors conducted in duplicates.

**Fig. 6 | In *Plasmodium falciparum*-infected pregnant women, opsonising antibodies against placental binding CS2-IEs correlate with protection from placental malaria.** Antibody-dependent phagocytosis (ADP) by (**a**) neutrophils and (**b**) monocytes of CS2-IEs opsonised using plasma from pregnant women diagnosed with malaria ($n = 77$) from Papua New Guinea is measured. Dots represent means from three experiments performed in duplicate using different whole blood donors, horizontal bars show group medians. ADP was expressed as relative phagocytosis, the percentage of the pooled immune plasma (PPS). Groups were compared using Mann Whitney-U Test. (Abbreviations: non-PM=non-placental malaria, PM=placental malaria, AU=Arbitrary Units).

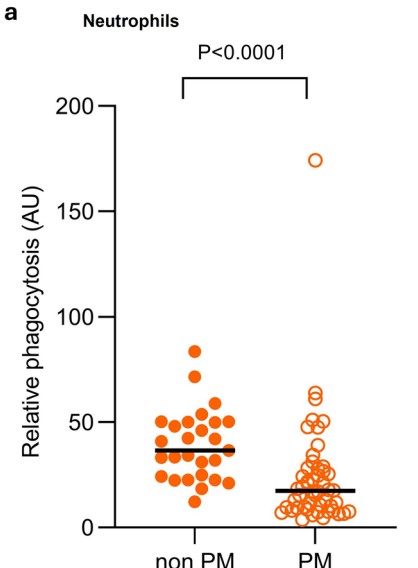
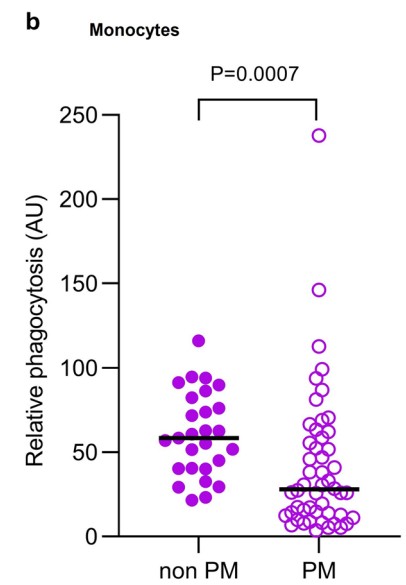

expression on neutrophils by mobilising intracellular pools[36]. Both TNFα and C5a are implicated in severe malaria and placental malaria[37–40]. TNFα and C5a showed dose-dependent induction of ADNP and ADCP, which was greatest when both were administered together. The benefits of increased clearance of IEs in the presence of these mediators may be outweighed by the induction of excessive inflammation either directly through neutrophil and monocyte activation or indirectly through effects on endothelial cells, platelets, and others.

PfEMP1 is the dominant VSA on the IE surface[8], and is a major target for opsonising antibodies measured using THP-1 cells[21]. In the whole blood assay, there appeared to be a component of non-PfEMP1-mediated opsonic phagocytosis. The SBP-1 knock-out line fails to export PfEMP1 to the surface, and using immune plasma, little IgG binding to IE was detected, but phagocytosis was still around 50% of that seen with CS2 IEs. This could represent an antibody to other IE surface antigens such as RIFIN, STEVOR, or GARP[41–43]. Phagocytosis of IE without PfEMP1 may also reflect recognition of altered host antigens on the IE surface[44] as shown by increased uptake of uninfected RBCs in the presence of BS3. The nature and identity of the non-PfEMP1 VSA require further characterisation.

Importantly, ADNP and ADCP correlated with protection from placental malaria, Antibodies to VAR2CSA are believed to play a critical role in protection against placental malaria[45]. A recent meta-analysis did not find strong evidence of associations between antibodies, primarily to recombinant VAR2CSA or individual protein domains, and protection[46], but a systems serology study of antibodies to VAR2CSA identified seven antibody features associated with protection from placental malaria, with functional antibodies that inhibit adhesion to chondroitin sulfate A or that relate to clearance of IE by neutrophils or THP-1 monocyte-like cells featuring

prominently[12]. In young Papua New Guinean children, ADNP using parasite line IT4VAR19 was substantially higher in children with uncomplicated malaria than in children with severe malaria. IT4VAR19 expresses a DC8-containing PfEMP1 and binds to EPCR, and its transcription has been associated with severe malaria[22,47], although it also does not bind to ICAM-1, a phenotype associated with cerebral malaria[11].

By contrast, ADCP of IT4VAR19 was similar in both groups on presentation and declined more substantially in convalescence in severe malaria. Kinetics of variant-specific antibody subclass responses following infection are highly variable[48] and differences in the importance of antibody isotypes or subclasses in ADNP and ADCP might explain some of these findings. Our whole blood assay provides strong evidence to support the role of functional antibodies to CS2 in protection from placental malaria and supports other findings that antibodies to VAR19 may protect against severe malaria[25].

Our assay has several important strengths. It allows the simultaneous measurement of antibodies that engage neutrophils and monocytes through their Fcγ receptors in a 96-well plate, high-throughput format, using small volumes of blood. Whole blood is diluted 1 in 4 but cells are not otherwise manipulated, providing a close approximation of the in vitro environment. Mature IEs reflect the parasite stages targeted by the immune response in vivo. The assay could readily be adapted to clinical settings, allowing a deeper understanding of how host immune cell activation may interact with circulating antibody to clear IEs, or fail to do so. Potential weaknesses include the current inability to assess NK cell responses in the same assay. Published assays of NK cell activation[32] by IE use a four-hour incubation and we have not detected NK cell activation (CD107a expression) at one hour in our phagocytosis assays (unpublished). Additionally, we were not able to

## Neutrophils

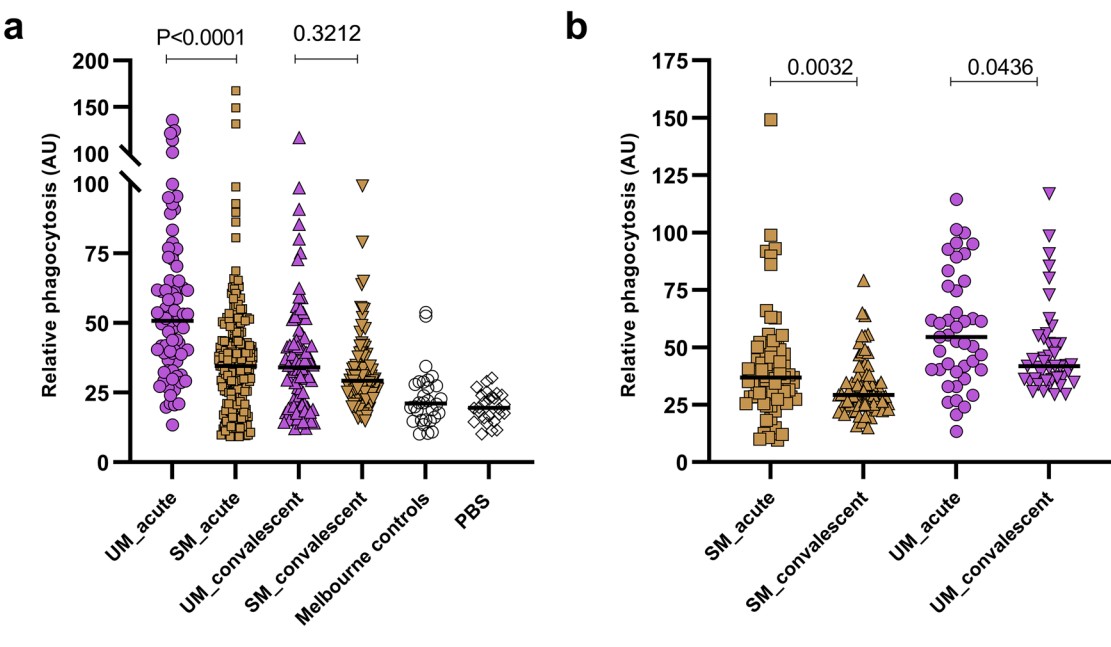

## Monocytes

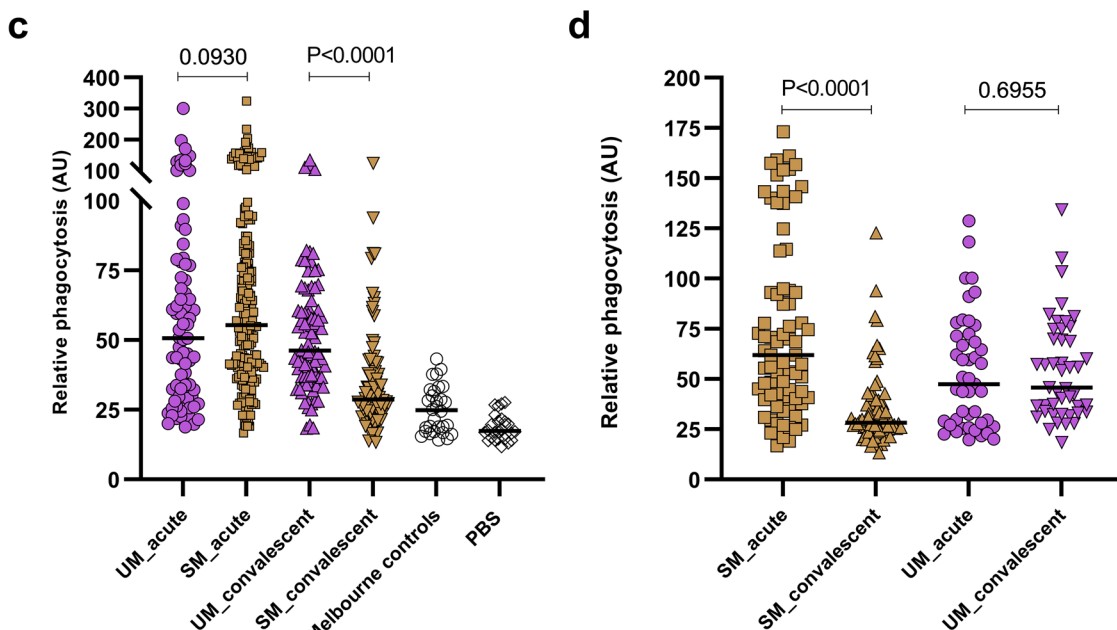

**Fig. 7 | Antibody-dependent phagocytosis of EPCR-binding IT4VAR19 IEs opsonised with plasma from children with malaria.** ADP of IEs by neutrophils (**a,b**) and monocytes (**c,d**) in whole blood was performed three times with different donors. ADP is represented as the mean of these three individual donors (dots) for each tested sample, and medians of each group are indicated with horizontal bars. Results are expressed as relative phagocytosis in arbitrary units (AU), the percentage of the pooled immune plasma (PPS). 462 plasma samples from children from PNG were tested; phosphate buffered saline (PBS) was included as a negative control. **a** and **c** uncomplicated malaria at presentation (UM_acute, n = 79), uncomplicated malaria in convalescence (UM_convalescent, *n* = 86), severe malaria at presentation (SM_acute, *n* = 211), and severe malaria in convalescence (SM convalescent, *n* = 86) children. Mann Whitney-U Test. **b** and **d** analysis of paired acute and convalescent plasma samples from 43 children with uncomplicated malaria and 75 children with severe malaria. Wilcoxon matched pairs rank test.

assess antibody responses to IE expressing a variety of PfEMP1 types in a single assay, which may be particularly important for severe malaria studies. The use of a DNA stain for IE precludes more detailed characterisation of non-parasite dependent phagocytosis. Future studies should aim to confirm

our clinical associations in other settings and to include a wider variety of parasite lines as targets, possibly in multiplex assays.

Whole blood phagocytosis assays are a practical tool for examining functional antibodies that contribute to ADNP and ADCP of *P. falciparum*

IEs, and may be important correlates of protection against malaria in pregnant women and young children.

## Data availability

The source data for Figs. 1–7 and Supplementary Figs. 1, 3, 4, 5 and 6 are included in Supplementary Data 1. Clinical data underlying Supplementary Table 2 are available in https://datadryad.org/dataset/doi:10.5061/dryad.wpzgmsbkx. The clinical data from all participants in the IMPROVE study and from Unger et al 2015 are available through the Worldwide Anti-malarial Resistance Network data repository on application. The clinical data from Papua New Guinean children are available from LM and ML on reasonable request.

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

## Acknowledgements

We acknowledge the support of Dr Darryl Johnson at the Image Stream facility and Dr Vanta Jameson from the Flow Cytometry core facility, both at the University of Melbourne. We thank the staff of the Papua New Guinea Institute of Medical Research, and the staff of the Paediatric Ward and Antenatal and Labour Wards of Modilon Hospital, Madang, for assistance with participant recruitment, and all participants and families for their contributions. In Malawi, we thank all the women who participated in the clinical trial that compared intermittent screening and treatment with intermittent preventive therapy and donated their plasma samples for this study, and all the physicians and nurses involved in the study. This work was supported by a National Health and Medical Research Council of Australia (NHMRC) grant to SJR and EHA (GNT1143946). The trial in pregnant Malawian women was supported by the European and Developing Countries Clinical Trial Partnership, Grant number: IP.2007.31080.003. Collection of samples from pregnant women in Papua New Guinea was supported by the Malaria in Pregnancy Consortium, through a grant from the Bill & Melinda Gates Foundation (46099); the Pregvax Consortium, through a grant from the European Union's Seventh Framework Programme FP7-2007-HEALTH (PREGVAX 201588); and Pfizer Inc (investigator-initiated research grant WS394663). Recruitment of participating children in Papua New Guinea was funded by the NHMRC (grant number: 513782).

## Author contributions

D.R., E.H.A. and SJR designed the study. D.R., W.H. and H.D. performed experiments and interpreted data. H.W.U. and M.O.-K. oversaw the pregnancy study and M.L. and L.M. ran the severe malaria study, both in Madang, Papua New Guinea. Ft.K. and M.M. oversaw the study in pregnant Malawian women. A.M. provided OmCI. B.W. and P.M.H. provided blocking antibodies to FcγRII. D.R. and S.J.R. drafted the paper and all authors approved the final manuscript for publication.

## Competing interests

The authors declare no competing interests.

## Additional information

¹Department of Medicine (RMH), Peter Doherty Institute of Infection and Immunity, University of Melbourne, Melbourne, Vic, Australia. ²Department of Infectious Diseases, Peter Doherty Institute of Infection and Immunity, University of Melbourne, Melbourne, Vic, Australia. ³Discovery Chemistry Research and Technologies, Eli Lilly & Co., Bracknell, UK. ⁴UWA Medical School, The University of Western Australia, Perth, WA, Australia. ⁵Papua New Guinea. Institute of Medical Research, Madang, Papua New Guinea. ⁶Global and Tropical Health Division, Menzies School of Health Research, Charles Darwin University, Casuarina, NSW, Australia. ⁷Department of Obstetrics and Gynaecology, Royal Darwin Hospital, Tiwi, NT, Australia. ⁸Department of Clinical Sciences, Liverpool School of Tropical Medicine, Liverpool, UK. ⁹Department of Clinical Sciences, Academy of Medical Sciences, Malawi University of Science and Technology, Thyolo, Malawi. ¹⁰Immune Therapies Group, Centre for Biomedical Research, Burnet Institute, Melbourne, Vic, Australia. ¹¹Department of Clinical Pathology, University of Melbourne, Melbourne, Vic, Australia. ¹²Department of Immunology and Pathology, Monash University, Melbourne, Vic, Australia. ¹³Department of Microbiology and Immunology, Peter Doherty Institute of Infection and Immunity, University of Melbourne, Melbourne, Vic, Australia. ✉e-mail: sroger@unimelb.edu.au

