## [Transparent Peer Review file · Communications Medicine]

A whole blood assay for antibody dependent phagocytosis of *Plasmodium falciparum* infected erythrocytes

Corresponding Author: Professor Stephen Rogerson

Version 0:

Reviewer comments:

Reviewer #1

(Remarks to the Author)

Dr. Rathnayake and colleagues describe a whole-blood assay to assess neutrophil and monocyte phagocytosis of IgG-opsonized *Plasmodium falciparum*-infected erythrocytes (IEs). The study is thorough and well-presented, and the assay will constitute a useful addition to the toolbox of malaria immunologists. The weakest part of the manuscript is the part that tries to assess the clinical relevance of the assay (see further below).

Specific comments:

=====

L. 49-52 (“ADNP and... severe malaria”): The authors state that ADNP but not ADCP was higher when IEs were opsonized by IgG from uncomplicated malaria than severe malaria. This somewhat surprising observation is not described or discussed in detail later in the manuscript.

L. 69-70 (“Malaria... 2020”): Please update using the latest version of the WHO World Malaria Report (2023).

L. 74-6 (“Defining... life cycle stages”): Why is studying the features (effector functions?) (as opposed to the targets) of IgG complicated by the complex life cycle of *P. falciparum*?

L. 76-9 (Uniquely,... dysfunction”): What is meant by “deep organs”? As far as I am aware, IEs sequester mainly in post-capillary venules, whether superficial or not.

L. 338-45 (“We hypothesized... about 5%”): The authors reasonably speculate on the involvement of non-PfEMP1 parasite antigens as targets of opsonizing IgG. However, they remain curiously silent about what they conclude from these experiments (that RIFIN-, STEVOR-, GARP(?) -specific IgG (targets which are not affected by the knockout) contributes significantly?).

L. 348-74 (“Seventy-seven... (Figure 7D)”): The authors relate the outcome of their assays to protection against placental malaria (PM) and severe malaria (SM). I suspect that the LEVELS of PfEMP1-specific IgG were higher, and the prevalence of PM was lower in multigravidae than in primigravidae. Thus, without correction for parity and/or levels, it appears to me that their assay merely provides a complicated measure of specific IgG levels (which could be assayed much simpler by ELISA). A similar argument applies to the SM vs. uncomplicated and acute vs. convalescence data. I suggest that the data are re-analyzed and corrected for (or as a minimum related to) differences in levels of the relevant IgG specificities. If this cannot be done, I suggest removing the parts of the manuscript relating the assay to outcome (in my opinion, the manuscript would still be valuable).

L. 352-4 (“Both... 0.69”): I suggest replacing “different donors” with “different effector cell donors” to remove confusion with samples from the plasma donors, which are also involved in this section.

L. 442-4 (“Potential weaknesses... severe malaria studies”): I am curious to know what the reason for the “current inability to assess NK cell responses in the same assay” might be. NK cells have recently received a lot of attention as potential effector cells against IgG-opsonized IEs (e.g., ref. 31).

Fig. 1A: I suggest clarifying the axis legends, to make it clearer what is "Blue SSC-A" and what is "Violet SSC-A)" [as described in L. 249-52].

Fig. 5C: Please provide a more elaborate figure legend, including symbols and lines (both dashed and solid).

Reviewer #2

(Remarks to the Author)

Rathnayake and colleagues propose a new test to measure Plasmodium phagocytosis in whole blood using a high-throughput approach. The approach is innovative in that it directly utilizes whole blood without purifying mononuclear cells, significantly reducing processing time. The tool has been validated across multiple populations with varying levels of immune response to Pf infection. The authors also targeted the major Pfemp1 antigen, which is crucial in the pathophysiology of infection, particularly focusing on a strain with placental adhesion phenotype and the IT4var19 strain associated with EPCR phenotype in cerebral malaria. Furthermore, the authors measured the dynamics of immune response in children from infection peak to convalescence.

To work directly on whole blood, it's necessary to lyse the red blood cells. The authors used FACs lysing solution directly. Could a pre-lysis step enhance mononuclear cell enrichment? For example, could specific lysis of red cells with streptolysin O (SLO) be beneficial? Have the authors tested any alternative lysis approaches?

The main difficulty lies in the quantity of available white blood cells, particularly for monocyte cells which are sparsely abundant in whole blood. The proportion of neutrophils and monocytes between the lysed and non-lysed unfixed conditions does not appear to be very similar. In the supplementary figure Fig1b, the neutrophil population (approximately 65-70%) is better represented compared to monocytes (10-15%). Could this favor ADNP?

In the discussion, the authors mention a higher yield of purified mononuclear populations but with the limitation of being removed from their original environment. Indeed, this is a significant advantage of the approach proposed here (lines 387-388). Do the authors have comparative data on ADNP, ADCP between the two approaches? To what extent can phagocytosis be influenced by cell quantity? If available, results comparing the two approaches could be added as supplemental data or more thoroughly discussed and documented in the discussion. Finally, has the viability of mononuclear cells been tested prior to the phagocytosis assay?

Another concern revolves around clarifying the number of replicates for the experiments and the nature of these replicates (biological and/or technical). Overall, graph interpretation would be facilitated by indicating the "n" value on each graph.

For example, in Figure 1D, it's not evident what each point represents (n=3). Is it 3 biological replicates or 3 different samples? This could be clarified in the legend. It's clear that statistical analyses with n=3 are not very robust; however, did the authors perform them? Could the number of replicates be increased for statistical analysis?

For this type of approach, does it make sense to test the reproducibility and sensitivity of the method? If so, do the authors have data that could be added as supplementary data?

The authors tested different concentrations of plasma on phagocytosis, and the range of dilution at which phagocytosis is observed appears similar between neutrophils and monocytes (1/2.5 and 1/5). However, the authors make a distinction in dilution between neutrophils and monocytes that is not clearly apparent in Figure 12a. Furthermore, overall, the number of engaged monocytes remains much lower than neutrophils. This was also observed in the initial manipulation between lysed and fixed cells, which seems to "favor" the number of neutrophils compared to other leukocyte cells. Was this expected between the two populations?

In Figure 2b, there doesn't appear to be a saturation plateau between the 1/5 and 1/2.5 dilutions in the PPS condition, unlike the Rah group. Conversely, for monocytes, there seems to be a beginning of a saturation plateau for the PPS group, similar to the positive control Rah. Have the authors tested with pure plasmas?

In Figure 2c, in the exposed group, there are some outlier points. Have the authors examined the malaria history of these women? Are these the same women found as outliers in both the neutrophil and monocyte groups? Ten plasma samples were randomly taken from infected and non-infected groups. In the infected group, was parasite density taken into account to have a panel of low, moderate, or high parasitemia in the group?

Figure 3: Do the two points correspond to two technical or biological replicates? The authors mention a decrease of approximately 50% in phagocytosis. With only two points, no statistical analysis is possible, but considering the data, it's unfortunate that the manipulation wasn't performed on a larger scale to enable statistical analysis of the observations.

The authors used a plasma pool from 50 women, and as they also had plasma from children who experienced uncomplicated or severe malaria, it would have been interesting to create plasma pools from these children for further investigations.

Figure 4: It needs clarification on what the measurements represent; it's unclear if the manipulation was performed in single, duplicate, or triplicate. Moreover, the standard deviation is significant for TNF α , which increases phagocytosis by 2-fold

(Figure 4a). In Figure 4b, the legend states that two independent experiments were conducted, but three points are observed on the graph. What do these correspond to? Have the authors conducted statistical analysis to measure the effect of TNF α vs. C5a or TNF α +C5a? Even though I agree that with n=3, this may be suboptimal. It's not easy to read the percentages described in the text on the graph; perhaps a dual-axis indicating the % of phagocytosis would aid in readability.

In Figure 5c, the authors could add "treated" and "untreated" to the legend.

In the section VI: did the authors investigate whether placental infection with Pf was recent or old, particularly through optical microscopy with the presence or absence of malarial pigment?

The authors do not observe any difference between the acute infection versus convalescence in the severe group for monocyte cells. Do they have a hypothesis or explanation for this result? Similarly, it is surprising not to observe differences between the UM and SM groups in the acute phase of infection.

In the discussion, the authors could qualify the assertion that ADNP and ADCP depend on plasma concentration and Pfemp1 expression (lines 379-380). This is because we cannot exclude that other VSA (Variant Surface Antigens) could play a role in the host immune response. The observations on the recognition of infected erythrocytes without Pfemp1 expression are interesting and support this idea.

Reviewer #3

(Remarks to the Author)

Rathnayake et al. describe a whole blood phagocytosis assay that represent an enhanced approach to existing assays that is anticipated to facilitate improved investigation into long-standing questions in the field of malaria around antibody functionality and phagocytosis. This assay does not require any cell extractions which is ideal for high throughput studies and allows measurement of infected-red blood cell phagocytosis by neutrophils and monocytes simultaneously. The authors have systematically investigated different aspects of the assay to establish complement contributions, and phagocytosis enhancement in presence of inflammatory molecules. Using this assay, the authors also measured antibody-dependent phagocytosis using samples from well-characterised cohorts which highlight the ability to discern an association between neutrophil and monocyte phagocytosis with malarial disease outcomes. While this is a well-written and easy to follow manuscript a few points need clarification:

1. Line 216-217 reads "CD45 was used as a standard membrane marker to distinguish DHE-labelled parasites within the cell membrane from parasites outside the membrane." Did the authors mean to say here that using the Amnis image stream system it was possible to distinguish DHE-labelled parasites within the cell membrane from parasites outside the membrane on CD45 expressing cells? If yes please edit to further clarify this. If not please include more detail of how CD45 could be used to determine intracellular versus extracellular RBCs.
2. Monocytes tend to become adherent when activated. Considering the relatively few cell counts of monocytes that were retrieved and analysed this may be important to consider. Can the authors please clarify if any detachment buffers were used to retrieve monocytes after incubation? If no detachment buffers or other means of detaching activated monocytes were performed, please demonstrate that it is not a necessary step to include in the protocol.
3. Please include clarification of how relative antibody units (RAU) are calculated/obtained.
4. Figure 5A. shows that antibodies from PPS plasma are deposited at a higher rate on uRBC than for MC plasma. This means that phagocytosis of IE may to some extent be parasite independent (as alluded to by the authors). As this assay precludes from using uninfected red blood cells as a background control (due to no DHE) to account for the non-parasite dependent phagocytosis, it should be noted in the limitations of the assay, at least when the aim is to assess parasite-antigen-specific antibody-dependent phagocytosis.
5. This assay measures phagocytosis by both neutrophils and monocytes from one sample. However, the relationship between these events within an individual is unclear? i.e. high neutrophil phagocytosis associated with high monocyte phagocytosis? Additionally, while important to understand the phagocytic contributions from each cell compartment, how does total/combined (both Neut+Mono) phagocytic capacity correlate to disease outcomes?

Version 1:

Reviewer comments:

Reviewer #1

(Remarks to the Author)

As far as I am concerned, Dr. Rathnayake and colleagues have adequately addressed most of the comments and concerns made by the reviewers. The most significant exception is their response to my comment regarding L. 348-74 in the original manuscript. I fully accept that the authors may not agree with my concern, which is possibly not justified. In their rebuttal, they refer to a series of earlier papers to build an indirect argument that indeed it is not. However, re-analysis of their current data with the proposed corrections would not be very onerous to do and would settle the matter directly. Why was this not done? Either way, the authors should add text to their manuscript regarding this issue, as it would be helpful to readers who may have concerns like mine.

For the same reason, the authors should add text to the manuscript regarding their observed differences in the response

kinetics of phagocytosis and ADCC as explained in their response to my comment to L. 442-4 of the original manuscript.

With respect to the comments of Reviewer 2, the new data provided in their response to the reviewer comment “The main difficulty... ADNP?” should be added to the manuscript as supplementary data, to alleviate reader concerns like those of the reviewer.

Finally, the histology data the authors mention in their response to the reviewer comment “In the section VI... pigment?” should be added to the manuscript text – again because it would be beneficial to readers of this interesting report.

Reviewer #2

(Remarks to the Author)

I would like to thank the authors for their re-analysis and corrections. The additional experiments to strengthen the work are much appreciated. The study is now sufficiently convincing and of interest for publication.

Reviewer #3

(Remarks to the Author)

The Authors have addressed all comments.

Reviewers' comments:

Reviewer #1 (Remarks to the Author):

Dr. Rathnayake and colleagues describe a whole-blood assay to assess neutrophil and monocyte phagocytosis of IgG-opsonized Plasmodium falciparum-infected erythrocytes (IEs). The study is thorough and well-presented, and the assay will constitute a useful addition to the toolbox of malaria immunologists. The weakest part of the manuscript is the part that tries to assess the clinical relevance of the assay (see further below).

Specific comments:

=====

L. 49-52 (“ADNP and... severe malaria”): The authors state that ADNP but not ADCP was higher when IEs were opsonized by IgG from uncomplicated malaria than severe malaria. This somewhat surprising observation is not described or discussed in detail later in the manuscript.

RESPONSE: Now addressed, Lines 423-6

L. 69-70 (“Malaria... 2020”): Please update using the latest version of the WHO World Malaria Report (2023).

RESPONSE: Updated (Lines 55-56)

L. 74-6 (“Defining... life cycle stages”): Why is studying the features (effector functions?) (as opposed to the targets) of IgG complicated by the complex life cycle of *P. falciparum*?

RESPONSE: We have added “including merozoites and infected erythrocytes, with distinct responses to each” (Line 63-64) as each stage is considered an important target of protective immunity, along with transitional processes such as schizont egress. The plethora of targets make detailed study of the features of any one response complex.

L. 76-9 (Uniquely,... dysfunction”): What is meant by “deep organs”? As far as I am aware, IEs sequester mainly in post-capillary venules, whether superficial or not.

RESPONSE: (Line 65) we changed this to “vasculature” given the universality of sequestration, predominantly in capillaries and post capillary venules.

L. 338-45 (“We hypothesized... about 5%”): The authors reasonably speculate on the involvement of non-PfEMP1 parasite antigens as targets of opsonizing IgG. However, they remain curiously silent about what they conclude from these experiments (that RIFIN-, STEVOR-, GARP(?)-specific IgG (targets which are not affected by the knockout) contributes significantly?).

RESPONSE: Text and references added in discussion, Lines 406-7.

L. 348-74 (“Seventy-seven... (Figure 7D)”): The authors relate the outcome of their assays to protection against placental malaria (PM) and severe malaria (SM). I suspect that the LEVELS of PfEMP1-specific IgG were higher, and the prevalence of PM was lower in multigravidae than in primigravidae. Thus, without correction for parity and/or levels, it appears to me that their assay merely provides a complicated measure of specific IgG levels (which could be assayed much simpler by ELISA. A similar argument applies to the SM vs. uncomplicated and acute vs. convalescence data. I suggest that the data are re-analyzed and corrected for (or as a minimum related to) differences in levels of the relevant IgG specificities. If this cannot be done, I suggest removing the parts of the manuscript relating the assay to outcome (in my opinion, the manuscript would still be valuable).

RESPONSE: We respectfully disagree with the reviewer. There is increasing evidence that measuring IgG to malaria antigens is rarely the best correlate of immunity {Suscovich, 2020 #165; Das, 2021 #467; Kurtovic, 2021 #468; Nziza, 2023 #469}. In relation to figure 6, we used the same cohort of pregnant women in a previously published “systems serology” study, including multiplex assays of IgG to VAR2CSA domains {Aitken, 2021 #243}. In that paper, IgG to VAR2CSA domains was not among the features that predicted protection from placental malaria in our machine learning-based analyses. The two groups were frequency matched for parity, to control for possible parity-dependent differences in antibody. (As the analyses are time-intensive we have not repeated the machine learning incorporating the new data).

In relation to figure 7, we previously published two papers examining IgG ELISA to PfEMP1 domains by ELISA or multiplex in a different subset of children with severe or uncomplicated malaria from the same study {Chan, 2019 #156; Rambhatla, 2019 #155}. In Chan et al, IgG to DBLalpha and DBLgamma domain on ITAR19 were higher in UM than SM, but antibody to CIDRalpha 1 and DBLbeta 12 were not. Similarly in Rambhatla et al, IgG antibodies to 35 CIDR domains did not differ at presentation between UM and SM. IgG antibodies to some CIDRalpha1 domains increased in convalescence, and this is the opposite of what we observed with the current assay, in which antibody to VAR19 IEs falls in convalescence for both ADNP and ADCP, most notably in severe malaria.

L. 352-4 (“Both... 0.69”): I suggest replacing “different donors” with “different effector cell donors” to remove confusion with samples from the plasma donors, which are also involved in this section.

RESPONSE: Change made

L. 442-4 (“Potential weaknesses... severe malaria studies”): I am curious to know what the reason for the “current inability to assess NK cell responses in the same assay” might be. NK cells have recently received a lot of attention as potential effector cells against IgG-opsonized IEs (e.g., ref. 31).

RESPONSE: NK cell activation occurs relatively more slowly than monocyte or neutrophil phagocytosis. We could not see any evidence of NK cell activation (CD107a expression increase) at 1 h in the current assay. Arora et al co-incubate NK

cells and infected erythrocytes for 4 h to measure degranulation.

Fig. 1A: I suggest clarifying the axis legends, to make it clearer what is “Blue SSC-A” and what is “Violet SSC-A” [as described in L. 249-52].

RESPONSE: We have modified the Y axis label and figure legend to address this

Fig. 5C: Please provide a more elaborate figure legend, including symbols and lines (both dashed and solid).

RESPONSE: We have added explanation of both the lines and the symbols used to the legend: “uRBCs were opsonised with rabbit antihuman erythrocyte IgGs (purple solid or dashed lines, open and closed circles), or pooled malaria naive plasma (black solid or dashed lines, open and closed squares), or were unopsonised (red solid or dashed lines, open and closed triangles).”

Reviewer #2 (Remarks to the Author):

Rathnayake and colleagues propose a new test to measure Plasmodium phagocytosis in whole blood using a high-throughput approach. The approach is innovative in that it directly utilizes whole blood without purifying mononuclear cells, significantly reducing processing time. The tool has been validated across multiple populations with varying levels of immune response to Pf infection. The authors also targeted the major Pfemp1 antigen, which is crucial in the pathophysiology of infection, particularly focusing on a strain with placental adhesion phenotype and the IT4var19 strain associated with EPCR phenotype in cerebral malaria. Furthermore, the authors measured the dynamics of immune response in children from infection peak to convalescence.

To work directly on whole blood, it's necessary to lyse the red blood cells. The authors used FACs lysing solution directly. Could a pre-lysis step enhance mononuclear cell enrichment? For example, could specific lysis of red cells with streptolysin O (SLO) be beneficial? Have the authors tested any alternative lysis approaches?

RESPONSE: We have not tested alternative approaches to lysis. Our preference is for minimal intervention with the whole blood prior to the assay, and standard lysis after phagocytosis is highly effective. In supplementary figure 1 we include comparisons of phagocytosis by neutrophils of IE in whole blood with and without lysis and/or fixation. (Because monocytes are significantly less abundant than neutrophils and acquisition of data are much slower we did not assess them in this experiment).

The main difficulty lies in the quantity of available white blood cells, particularly for monocyte cells which are sparsely abundant in whole blood. The proportion of neutrophils and monocytes between the lysed fixed and non-lysed unfixed conditions does not appear to be very similar. In the supplementary figure Fig1b, the neutrophil population (approximately 65-70%) is better represented compared to monocytes (10-15%). Could this favor ADNP?

RESPONSE: We did not actually include assessments of monocyte phagocytosis in those early experiments in Supp Fig 1. In Supp Fig 1 B we show the relative proportions of neutrophils, monocytes, lymphocytes and basophils/eosinophils in four different donors when cells had been lysed and fixed. The proportions of each cell type are in accordance with their expected proportions in intact blood.

Figure 1 D and E compare the numbers of IE phagocytosed by neutrophils and by monocytes (D) and the proportions of neutrophils and monocytes engaged in phagocytosis (E). The variation in phagocytosis is driven by variation in the original numbers of the different leukocyte types. We agree that our findings would suggest that ADNP may be more clinically important in vivo given the greater relative abundance of neutrophils.

We performed experiments to address the question of possible disproportionate loss of monocytes (below). We measured monocyte and neutrophil counts in (1) “fresh blood” diluted with culture medium without infected erythrocytes (2) “no treatment”, opsonic phagocytosis assays performed using our usual protocol (3) “1%BSA/EDTA”, assays performed as usual, but with plates blocked with 1% BSA and with 2 mM EDTA added (figure below).

There was some decrease in cell counts for both cell types when incubated with positive control (rabbit antihuman RBC) or negative control (non- opsonised RBC) but the decrease was similar for both monocytes and neutrophils, with similar differences between positive and negative controls. Thus, we do not see a disproportionate loss of monocytes.

In the discussion, the authors mention a higher yield of purified mononuclear populations but with the limitation of being removed from their original environment. Indeed, this is a significant advantage of the approach proposed here (lines 387-388). Do the authors have comparative data on ADNP, ADCP between the two approaches?

To what extent can phagocytosis be influenced by cell quantity? If available, results comparing the two approaches could be added as supplemental data or more thoroughly discussed and documented in the discussion.

RESPONSE: We have added Supplementary table 4 which correlates ADNP as measured with isolated cells (using previously published data from {Aitken, 2021 #243}) with neutrophil phagocytosis measured in whole blood with the same samples (data from Figure 6), and similarly for monocytes with ADCP by purified monocytes. Neutrophil and monocyte phagocytosis with purified effectors are moderately correlated and results with the whole blood assay are strongly correlated for neutrophils and monocytes in the same assay, but the correlation between the two assay types is not significant, as described in lines 342-4. The complement data suggest this is an important factor in the difference between the ADNP with purified cells and with whole blood.

Finally, has the viability of mononuclear cells been tested prior to the phagocytosis assay?

RESPONSE: In all the current experiments less than 30 minutes elapsed between collecting the blood and adding the opsonised infected erythrocytes, so we have not specifically tested cell viability.

Another concern revolves around clarifying the number of replicates for the experiments and the nature of these replicates (biological and/or technical). Overall, graph interpretation would be facilitated by indicating the "n" value on each graph.

For example, in Figure 1D, it's not evident what each point represents (n=3). Is it 3 biological replicates or 3 different samples? This could be clarified in the legend. It's clear that statistical analyses with n=3 are not very robust; however, did the authors perform them? Could the number of replicates be increased for statistical analysis?

RESPONSE: We have reworded the figure legends for clarity. Samples were run in duplicate for each experiment, and the mean of duplicates is plotted in Figure 1 D and E. Three different whole blood donors were used. Where relevant, each figure legend contains similar data, all points reflect means from individual experiments.

For this type of approach, does it make sense to test the reproducibility and sensitivity of the method? If so, do the authors have data that could be added as supplementary data?

RESPONSE: We think the most useful way of examining this is in the inter-donor variation shown in Figure 2 D, in which 40 samples were used as opsonins for three different donors (each tested in duplicate).

The authors tested different concentrations of plasma on phagocytosis, and the range of dilution at which phagocytosis is observed appears similar between neutrophils and monocytes (1/2.5 and 1/5). However, the authors make a distinction in dilution between neutrophils and monocytes that is not clearly apparent in Figure 2a. Furthermore, overall, the number of engaged monocytes remains much lower than neutrophils. This was also observed in the initial manipulation between lysed

and fixed cells, which seems to "favor" the number of neutrophils compared to other leukocyte cells. Was this expected between the two populations?

RESPONSE: As mentioned elsewhere (page 6) we only examined neutrophil phagocytosis in Supplementary Figure 1, as well as quantitating relative percentages of the different cell types after lysing and fixing cells.

In Figure 2 a, we are reporting (1) total numbers of phagocytic monocytes and neutrophils (left axis) and monocytes and neutrophils as percentages of the total phagocytic cell population (right axis). The percentages of phagocytic cells that are neutrophils vs monocytes is broadly consistent across plasma concentrations, with neutrophils being 80-90% of phagocytic cells which is in keeping with their being significantly more abundant than monocytes in whole blood, around 40-60% vs 5-8% of leukocytes.

In Figure 2b, there doesn't appear to be a saturation plateau between the 1/5 and 1/2.5 dilutions in the PPS condition, unlike the Rah group. Conversely, for monocytes, there seems to be a beginning of a saturation plateau for the PPS group, similar to the positive control Rah. Have the authors tested with pure plasmas?

RESPONSE: We did not test undiluted plasma. It was not very practical to do so.

In Figure 2c, in the exposed group, there are some outlier points. Have the authors examined the malaria history of these women? Are these the same women found as outliers in both the neutrophil and monocyte groups?

RESPONSE: Yes, in Supplementary figure 4 these data are plotted against one another. The outliers correlate quite closely. We have referenced this figure in the text highlight this point (line 286).

Ten plasma samples were randomly taken from infected and non-infected groups. In the infected group, was parasite density taken into account to have a panel of low, moderate, or high parasitemia in the group?

RESPONSE: We did not control for parasitaemia given the small group sizes. Being semi-immune pregnant women, parasite densities are often relatively low unlike in children, and the relationship between antibody and parasitaemia is not very clear.

Figure 3: Do the two points correspond to two technical or biological replicates? The authors mention a decrease of approximately 50% in phagocytosis. With only two points, no statistical analysis is possible, but considering the data, it's unfortunate that the manipulation wasn't performed on a larger scale to enable statistical analysis of the observations.

RESPONSE: We have replaced figure 3 with entirely new data. We needed to use higher concentration of the complement inhibitors in the new experiments to reproducibly inhibit phagocytosis, and have omitted the heat inactivated plasma condition as separating the plasma and cells often resulted in activation. Three experiments were performed three times in duplicate. Differences are not statistically significant due to sample size, and we report percentage differences.

The authors used a plasma pool from 50 women, and as they also had plasma from children who experienced uncomplicated or severe malaria, it would have been interesting to create plasma pools from these children for further investigations.

RESPONSE: Thank you, the small volumes of plasma remaining from this cohort precluded doing this, but this would be a valuable approach in our ongoing studies on a new cohort.

Figure 4: It needs clarification on what the measurements represent; it's unclear if the manipulation was performed in single, duplicate, or triplicate. Moreover, the standard deviation is significant for TNF α , which increases phagocytosis by 2-fold (Figure 4a). In Figure 4b, the legend states that two independent experiments were conducted, but three points are observed on the graph. What do these correspond to? Have the authors conducted statistical analysis to measure the effect of TNF α vs. C5a or TNF α +C5a? Even though I agree that with n=3, this may be suboptimal. It's not easy to read the percentages described in the text on the graph; perhaps a dual-axis indicating the % of phagocytosis would aid in readability.

RESPONSE: We present results of three independent experiments performed in duplicate. Dots represent means from each experiment. We have edited the legend to reflect this. We did perform statistical analysis which did not reveal statistically significant differences.

In Figure 5c, the authors could add "treated" and "untreated" to the legend.

RESPONSE: (See also reviewer 1 response). We have revised the legend to make it clearer, including defining the symbols

In the section VI: did the authors investigate whether placental infection with Pf was recent or old, particularly through optical microscopy with the presence or absence of malarial pigment?

RESPONSE: Of the 50 women with current infection on histology. 20 had "acute" infection (parasites only) and 30 had chronic (parasites plus pigment in monocytes +/- fibrin).

Of 27 women with no active infection at delivery, 12 had past infection (pigment only) and 15 had no infection on histology.

The authors do not observe any difference between the acute infection versus convalescence in the severe group for monocyte cells. Do they have a hypothesis or explanation for this result? Similarly, it is surprising not to observe differences between the UM and SM groups in the acute phase of infection.

RESPONSE: We agree the neutrophil responses are more in keeping with possible expectations. It is clear there is great heterogeneity in the kinetics of antibody isotype responses to IE VSA following infection {Kinyanjui, 2003 #473}, and this may partly explain the lack of association. Given the involvement of complement, dissecting out the roles of Ig isotypes or subclasses is difficult. There was a marked decline in opsonising activity for monocytes in SM between presentation and convalescence

and this may be an important factor, suggesting a boosting of antibody early in infection before presentation, and rapid decline in convalescence. Pre-infection samples, if they had been available, could have helped clarify that. We've discussed these differences (Lines 424-426).

In the discussion, the authors could qualify the assertion that ADNP and ADCP depend on plasma concentration and Pfemp1 expression (lines 379-380). This is because we cannot exclude that other VSA (Variant Surface Antigens) could play a role in the host immune response. The observations on the recognition of infected erythrocytes without Pfemp1 expression are interesting and support this idea.

RESPONSE: Thank you, we have added to the discussion (lines 406-7).

Reviewer #3 (Remarks to the Author):

Rathnayake et al. describe a whole blood phagocytosis assay that represent an enhanced approach to existing assays that is anticipated to facilitate improved investigation into long-standing questions in the field of malaria around antibody functionality and phagocytosis. This assay does not require any cell extractions which is ideal for high throughput studies and allows measurement of infected-red blood cell phagocytosis by neutrophils and monocytes simultaneously. The authors have systematically investigated different aspects of the assay to establish complement contributions, and phagocytosis enhancement in presence of inflammatory molecules. Using this assay, the authors also measured antibody-dependent phagocytosis using samples from well-characterised cohorts which highlight the ability to discern an association between neutrophil and monocyte phagocytosis with malarial disease outcomes. While this is a well-written and easy to follow manuscript a few points need clarification:

1. Line 216-217 reads "CD45 was used as a standard membrane marker to distinguish DHE-labelled parasites within the cell membrane from parasites outside the membrane." Did the authors mean to say here that using the Amnis image stream system it was possible to distinguish DHE-labelled parasites within the cell membrane from parasites outside the membrane on CD45 expressing cells? If yes please edit to further clarify this. If not please include more detail of how CD45 could be used to determine intracellular versus extracellular RBCs.

RESPONSE: Apologies for the lack of clarity, it is the former, we have edited the text (lines 208-9)

2. Monocytes tend to become adherent when activated. Considering the relatively few cell counts of monocytes that were retrieved and analysed this may be important to consider. Can the authors please clarify if any detachment buffers were used to retrieve monocytes after incubation? If no detachment buffers or other means of detaching activated monocytes were performed, please demonstrate that it is not a necessary step to include in the protocol.

RESPONSE: See response to reviewer 2, page 6 and included figure. We found no evidence of selective depletion of monocytes due to activation.

3. Please include clarification of how relative antibody units (RAU) are calculated/obtained.

RESPONSE: In Figure 2 c, Figure 6 and Figure 7, relative antibody units are expressed as percentages relative to our positive plasma pool, after subtracting no-plasma controls from each. This was stated in the legends for Figures 2 c and 6, and has been added to the legend for Figure 7 (Lines 564-5).

In Figure 3 and 4 the Y axis and legend now indicate the data are phagocytosis, relative to the control.

4. Figure 5A. shows that antibodies from PPS plasma are deposited at a higher rate on uRBC than for MC plasma. This means that phagocytosis of IE may to some extent be parasite independent (as alluded to by the authors). As this assay precludes from using uninfected red blood cells as a background control (due to no DHE) to account for the non-parasite dependent phagocytosis, it should be noted in the limitations of the assay, at least when the aim is to assess parasite-antigen-specific antibody-dependent phagocytosis.

RESPONSE: Thank you, we have added this caveat to the discussion (Lines 440-1).

5. This assay measures phagocytosis by both neutrophils and monocytes from one sample. However, the relationship between these events within an individual is unclear? i.e. high neutrophil phagocytosis associated with high monocyte phagocytosis? Additionally, while important to understand the phagocytic contributions from each cell compartment, how does total/combined (both Neut+Mono) phagocytic capacity correlate to disease outcomes?

RESPONSE: Supplementary Figure 4 and the Supplementary Table address the relationship between the effector cells, correlating the neutrophil phagocytosis and monocyte phagocytosis in the 40 samples evaluated in Figure 2. There is a moderate to strong correlation between the two. The outliers with high antibody correlate well.

We don't feel that a combined neutrophil and monocyte uptake would be very helpful. The ratio of the effectors varies donor to donor and day to day and monocytes are usually outnumbered 10:1 or so by neutrophils. Based on this and our data, the latter would appear likely to have the dominant role.

Nov 2024 response to reviewers

Reviewer #1 (Remarks to the Author):

As far as I am concerned, Dr. Rathnayake and colleagues have adequately addressed most of the comments and concerns made by the reviewers.

The most significant exception is their response to my comment regarding L. 348-74 in the original manuscript. I fully accept that the authors may not agree with my concern, which is possibly not justified. In their rebuttal, they refer to a series of earlier papers to build an indirect argument that indeed it is not. However, re-analysis of their current data with the proposed corrections would not be very onerous to do and would settle the matter directly. Why was this not done? Either way, the authors should add text to their manuscript regarding this issue, as it would be helpful to readers who may have concerns like mine.

RESPONSE: We have added a new Supplementary Figure 6 (text lines 357-358), showing that neither ADNP nor ADCP correlate with IgG to full length FCR3 VAR2CSA. This seems a more convincing way of showing that ADNP and ADCP provide data that does not “merely provides a complicated measure of specific IgG levels (which could be assayed much simpler by ELISA)” as the reviewer’s original comments stated. We have also added methods for the ELISA (lines 225-232).

For the same reason, the authors should add text to the manuscript regarding their observed differences in the response kinetics of phagocytosis and ADCC as explained in their response to my comment to L. 442-4 of the original manuscript.

RESPONSE: We have added more detail to the “weaknesses” of the assay (Line 452-454). Both reference 32 (Arora et al), using IEs, and papers by Larsen et al Nat Comms 2021 (plate based NK activation by opsonised antigens) use 4 h incubation.

With respect to the comments of Reviewer 2, the new data provided in their response to the reviewer comment “The main difficulty... ADNP?” should be added to the manuscript as supplementary data, to alleviate reader concerns like those of the reviewer.

RESPONSE: We have added these data as Supplementary Figure 3 (and text lines 270-271).

Finally, the histology data the authors mention in their response to the reviewer comment “In the section VI... pigment?” should be added to the manuscript text – again because it would be beneficial to readers of this interesting report.

RESPONSE: We have added these data to Supplementary Table S2.